# HETEROGENEITY OF REGULARIZATION BETWEEN ADJACENT PERIODS

## ABSTRACT

Since the inception of deep learning, regularization techniques have been developed for the purpose of preventing the overfitting phenomenon Ying (2019b). RegularizationKukačka et al. (2017) is typically accomplished in two ways: incorporating randomnessScardapane & Wang (2017) (e.g., injecting noise into data, activating nodes, or using dropoutAngermueller et al. (2016)) or heterogeneity (e.g., data augmentationShorten & Khoshgoftaar (2019)). These approaches are known to lead to better generalizationFan et al. (2021) and, consequently, improved performance. In the case of introducing heterogeneity by adjusting the hyperparameter during the training processAndonie (2019), such as the drop rate of dropout, experiments have shown that tuning hyperparameters after a period, which consists of a certain number of forward propagations, is more effective than either uniformly sustaining hyperparameters or tuning them during every propagation. Therefore, this paper proposes a novel regularization technique named Periodic Regularization that introduces periodicity into the dynamic hyperparameter tuning of other regularization methods. Furthermore, this paper suggests combining Periodic Regularization and other learning techniques such as Reinforcement Learning (RL)Jaafra et al. (2019) and Transfer Learning Weiss et al. (2016). This approach, particularly when combining dropout and Reinforcement Learning, shows significant improvement in empirical testing across various popular datasets. This is notably evident in Facial Expression Recognition (FER) tasks Li & Deng (2020), where conventional methods, such as noise injectionNoh et al. (2017) and dropout, have proven ineffective. Our proposed periodic regularization method not only can fill the research gap found in traditional regularization techniques but also can be a cornerstone for further research where the concept of periodic regularization is combined with diverse vanilla regularization techniques and learning techniques.

## 1 INTRODUCTION

Deep learning has made noticeable advancements in various fields, most notably in computer vision. The revolutionary improvements are evident in classification tasks across datasets such as CIFAR100 and Facial Expression Recognition (FER) datasets, such as FER2013 Giannopoulos et al. (2018)and Affect Net Mollahosseini et al. (2019).

Although deep learning has made models more robust than before, improving performance remains challenging due to overfitting. Overfitting occurs when a deep learning model extensively memorizes the noise in a training dataset, decreasing its performance on validation or test datasets.

To alleviate overfitting, regularization techniques are employed during the training phase. Regularization aims to enhance a model's generalization by introducing constraints during training. This can be performed through direct modification of the objective function, as seen in L1 and L2 regularizationYing (2019a). Another approach is to inject randomness such as noise injection and dropout and to increase heterogeneity through methods like data augmentation.

However, when applying regularization, several challenges also arise. First, aggressive utilization of regularization can rather lead to performance degradation. Second, hyperparameter tuning is highly laborious. Furthermore, it is challenging to choose hyperparameters appropriate at specific moments

during training. Third, there's inconsistency for improving performance of model, depending on the type of dataset.

Despite these limitations, utilizing regularization is still crucial for enhancing model performance. This is especially important in domains prone to overfitting due to excessive noise in the data Mollahosseini et al. (2019). Thus, efforts have been made to develop regularization techniques that both consistently improve the performance regardless of domain and can be applied cost-effectively. These attempts have been conducted by adding or combining various "elements" with existing regularization methods. As an representative example adding heterogeneity, there is an augmentation that applies a variety of heterogeneous modifications to the input layer, prompting the model to learn general features in the data.

On the other hand, to increase "heterogeneity between adjacent propagations" at the hidden layer level, dynamically adjusting hyperparameters during training can be devised, as even minor changes can have a significant heterogeneity during training. For example, the noise injection, whose the standard deviation in the Gaussian distribution dynamically changes every forward propagation ,can be devised.

Given that this noise injection results in a range of heterogeneous forward propagations, it can be anticipated that applying this algorithm can enhance the model's robustness, akin to data augmentation. However, contrary to expectations, introducing heterogeneity in the hidden layers actually degraded performance while data augmentation in the input layers improves performance.

To address this issue, our paper introduces, as its first contribution, the concept of Periodic Regularization. This concept dynamically alters hyperparameters for each "period"[1], comprised of multiple forward propagations, allowing the model to adapt to the training data with sufficient time. Furthermore, it is shown that the model's performance can be more robust with applying regularization which "extends" Jackson & Jackson (2014)[1] the concept of periodic regularization than with vanilla regularization.

Furthermore, as primary contribution, this paper presents the innovative methodology called "task transformation"[1]. This idea is rooted in characteristics of periodic regularization—such as sequential decision problemsPuterman (1990) related to hyperparameters, heterogeneity, and period length—all of which are discussed in the method section. This term, task transformation, means that the pre-transformed task whose goal is to train the model by dynamically changing the hyperparameters can be transformed into a different task such as the MDP Puterman (1990) problem[1] and transfer learning problem [1] due to traits of periodic regularization.

By transforming the task into those different tasks, solutions, such as RL and fine-tuning Paul et al. (2015)Jeddi et al. (2020) which are originally devised to address the MDP and transfer learning problems respectively, can be applied to solve the pre-transformed task [1] for further improvement. In short, the regularization building upon the concept of periodic regularization can apply task transformation. As a result, training techniques like RL or fine-tuning can be "implemented" Burdy et al. (2003)[1]to enhance performance.

To showcase the superiority of utilizing RL and transfer learning for addressing the pre-transformed task, we solve the aforementioned problems that many previously devised regularizations struggling with. In other words, this paper aims to show that Periodic regularization, when implementing "RL"[1], not only consistently can enhance performance across different datasets, but also automatically can adjust appropriate hyperparameters with regard to the state of parameters, effectively preventing overfitting. Moreover, it is also shown that utilizing "fine-tuning" [1] can allow the model to exploit advantages of transfer learning 2.3 even with a single dataset.

Specifically, this paper tackles the tricky problem where representative regularization techniques such as noise injection and dropout cannot make meaningful improvements on the FER dataset such as FER2013 and Affect Net while those have noticeable effectiveness on the well-known dataset such as CIFAR100. Through addressing this challenging problem, beyond comparing performance from a numerical perspective, we demonstrate how well the proposed method can represent the features of the data by utilizing grad CAM Selvaraju et al. (2017).

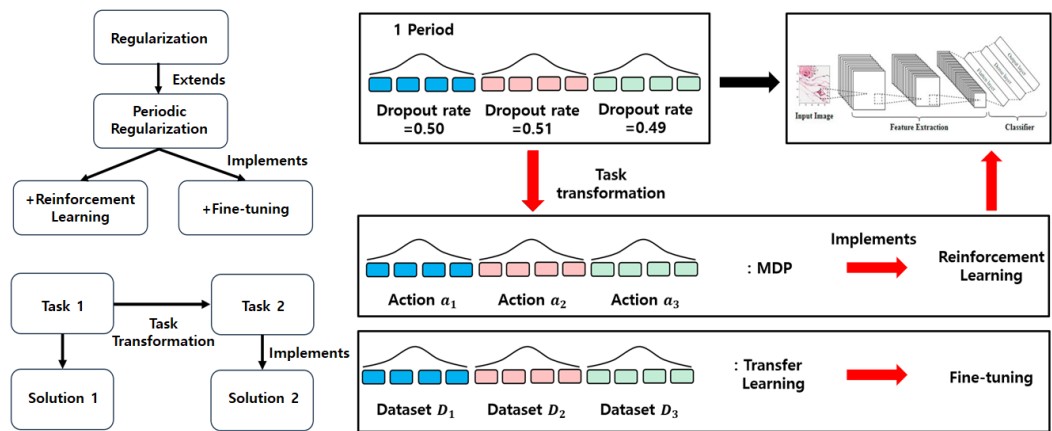

Figure 1: Relationship among Periodic regularization, Extends, Task transformation, Implements

## 2    RELATED WORKS

### 2.1    LOSS LANDSCAPE AND REGULARIZATION

The loss landscape Li et al. (2018) is a tool to graphically represent the losses generated through the training process. Moreover, given that heterogeneous regularization techniques impose unique constraints or penalties on parameters, the regularization influence trajectory of generated losses, leading to modification of loss landscape. Therefore, applying heterogeneous regularizations can make distinct loss landscapes3. Kunin et al. (2019).

### 2.2    REINFORCEMENT LEARNING AND PPO

Reinforcement Learning (RL) teaches agents to optimize decisions sequentially for maximum rewards. To apply RL, Markov Decision Processes (MDPs), encompassing states, actions, transition probabilities, and rewards, must be defined. However, due to MDP challenges for transition probabilities, model-free Liu et al. (2021) techniques is usually utilized. For the type of RL algorithm, there are two primary RL training methods Arulkumaran et al. (2017): value-based, such as Q-learning Arulkumaran et al. (2017), and policy-based. A notable advanced RL algorithm is Proximal Policy Optimization (PPO) Engstrom et al. (2020). The PPO employs a clipped objective function, which acts as a safeguard against drastic policy changes. proving effective in diverse tasks.

To narrow down only to policy-based agents, especially PPO, terms are denoted as follow: a $\pi$ is a policy utilized when the agent decide actions what. $\theta$ represents the parameter of a policy $\pi_\theta$, and a trajectory $\tau$ is a sequence of states and actions: $\tau = (s_1, a_1, \ldots, s_n, a_n)$. The return is the cumulative reward over a trajectory: $R(s) = \sum_{t=1}^{n} r_t$. To present action in advance, the hyperparameters are configured as action when the RL is leveraged with proposed method below.

The goal of the agent is to find the $\pi$ that can maximize the expected return: $\max_\theta J(\pi_\theta) = \arg\max_\theta \mathbb{E}_{\tau \sim \pi_\theta} [R(\tau)]$. To achieve this, a gradient ascent algorithm $\theta_{k+1} = \theta_k + \alpha \nabla_{\theta_k} J(\pi_{\theta_k})$ is utilized. Specifically, PPO-clip, is utilized in this paper. The objective function of the PPO is defined by: $\max_\theta J(\pi_\theta) = \arg\max_\theta \mathbb{E}_{s,a \sim \pi_{\theta_k}} [L(s, a, \theta_k, \theta)]$ where $L$ is given by $L(s, a, \theta_k, \theta) = \min\left(\frac{\pi_\theta(a|s)}{\pi_{\theta_k}(a|s)} A^{\pi_{\theta_k}}(s,a), \text{clip}\left(\frac{\pi_\theta(a|s)}{\pi_{\theta_k}(a|s)}, 1-\epsilon, 1+\epsilon\right) A^{\pi_{\theta_k}}(s,a)\right)$.

$A$, the advantage function, is defined as $A^{\pi_{\theta_k}} = R(s_k) - b$. In this paper, $R(s_k)$ is denoted as the difference between the mean of test F1 score and training F1 score of the $k^{th}$ sample and $b$, baseline function, is predicted by the MLP. Moreover, $\epsilon$ is configured $\epsilon = 0.2$.Wu et al. (2020).

## 2.3 TRANSFER LEARNING AND FINE-TUNING

Transfer learning is to pass down the previously trained parameters for one task to secondary related task. In this process, the parameters are adjusted according to the secondary task, which this is called fine-tuning. This approach can conserves time consumed for training because the previously trained parameters can facilitate the training to new data. Moreover, It can help to mitigate overfitting because the parameters are accumulatively trained on the various data.

## 2.4 FER DATASET

Tasks in FER(Facial Expression Recognition) are struggling to overfitting because of the nature of the domain where there is a lot of noise in data Li & Deng (2020). Furthermore, conventional methods like Gaussian noise injection or dropout aren't always effective in FERXie et al. (2019).

## 2.5 GRAD-CAM

Grad-CAM, or Gradient-weighted Class Activation Mapping, helps in this by highlighting key image areas that influence model classifications 5. It evaluates gradients in the final convolutional layer to produce a localization map. In order to obtain the class-discriminative localization map ($L_{\text{Grad-CAM}}^c \in \mathbb{R}^{u \times v}$ of width $u$ and height $v$ for any class $c$), it is necessary to compute the gradient of the score for class $c$, $y_c$ (before the softmax), with respect to feature maps $A_k$ of a convolutional layer ($\frac{\partial y_c}{\partial A_k}$). These gradients flowing back are global-average-pooled to obtain the neuron importance weights $\alpha_k^c$: $\alpha_k^c = \frac{1}{Z} \sum_i \sum_j \frac{\partial y_c}{\partial A_{k_{ij}}} \frac{\partial y_c}{\partial A_k}$. This weight $\alpha_k^c$ captures the importance of feature map $k$ for a target class $c$. Lastly, insert it into ReLU to filter the only positive relationship: $L_{\text{Grad-CAM}}^c = \text{ReLU} \left( \sum_k \alpha_k^c A_k \right)$Selvaraju et al. (2017).

# 3 METHOD

In this section, we present the definition of periodic regularization, factors that should be considered when utilizing periodic regularization, and introduce techniques such as periodic noise injection and periodic dropout that extend the concept of periodic regularization. Moreover, the justification about why the regularization extending the periodic regularization can be transformed into previously presented problems such as MDP and transfer learning. Also, how RL and fine-tune are actually implemented into periodic regularization is introduced.

## 3.1 DEFINITION OF PERIODIC REGULARIZATION

Periodic regularization is defined as a sequence of different regularizations, each with its unique hyperparameters which are exclusively applied at certain a period. As important property of periodic regularization, there are length of period and degree of heterogeneity between adjacent periods. The period's length consists of the number of forward propagations, and the regularization with its own hyperparameters shifts as each period elapses, generating heterogeneity. Therefore, this methodology can introduce heterogeneity at the hidden layer level. This heterogeneity can enhance the model's adaptability and generalization.

## 3.2 FACTORS THAT SHOULD BE CONSIDERED FOR EFFECTIVE APPLICATION OF PERIODIC REGULARIZATION

Successfully leveraging periodic regularization requires a balanced consideration of both the heterogeneity among regularization techniques and the length of periods.

First, if the heterogeneity among regularization techniques is too subtle, the difference between performance improvements achieved with periodic regularization and the original regularization might be indistinguishable. Conversely, if the heterogeneity is too big with short length of periods, the model can be underfitted, which can be confirmed through the peak in train losses3. To deal with this phenomenon, the length of period should be big enough to let the parameter adapt to the data.

Furthermore, it's crucial to determine the appropriate period's length. This is because, even with the right degree of heterogeneity, overly frequent shifts due to short period lengths can disrupt the model's ability to cumulatively learn, fit, or adapt to the data. This is the cause of underperformance observed in previous noise injection techniques with a single propagation.

### 3.3 PERIODIC REGULARIZATION AS AN ABSTRACT CLASS

Conceptually, periodic regularization is not a standalone regularization method like noise injection or dropout. It's akin to an abstract class in JavaSabharwal (1998), where existing methods like noise injection and dropout extend periodic regularization1. Below we introduce how noise injection and dropout extend the concept of periodic regularization, respectively named periodic Gaussian noise injection and periodic dropout1.

#### 3.3.1 PERIODIC GAUSSIAN NOISE INJECTION

Noise injection is one of the representative regularization techniques that injects random noise into activated nodes in a hidden layer. The noise is sampled usually from the Gaussian distribution. In the case of general noise injection, all individual noise injection layers sample the noise from the Gaussian distribution with static mean and standard deviation. However, in the case of the Gaussian noise injection extending periodic regularization 1, the standard deviation of each injection layer periodically changes, increasing the heterogeneity during the training process. This increased heterogeneity can improve the performance of the model.

#### 3.3.2 PERIODIC DROPOUT

Dropout is a regularization technique that drops certain nodes in the hidden layer. Usually, the drop rate, the hyperparameter of dropout, is sustained uniformly. However, in the case of the dropout extending periodic regularization 1, all individual dropout layers change the drop rate on each period. This leads to increasing the heterogeneity during the training process, increasing the performance of the model.

As illustrated above, one advantage of periodic regularization can be to enhance the performance of standard regularization by inheriting the properties of periodic regularization. However, the most salient trait of periodic regularization is the extensibility because various existing regularizations can also extend the concept with just simple modification from vanilla regularization techniques.

#### 3.3.3 WHY PERIODIC REGULARIZATION CAN BE TRANSFORMED INTO DIFFERENT TASK

In the context of periodic regularization, hyperparameters are determined sequentially for each period. Therefore, this can be viewed as a sequential decision problem for hyperparameters. However, randomly selecting them may not ensure optimal outcomes against overfitting. Although there are some techniques adjusting hyperparameters during training based on backpropagation Hataya et al. (2020), they focus on minimizing training loss, not test loss.

Considering the characteristics of the problem, the hyperparameter decision process for each period can be viewed as an MDP problem where state is defined as model's current status and the action is defined as deciding the hyperparameter. Moreover, if RL is utilized to train the child modelBello et al. (2017), model's performance can be improved by deciding the optimal hyperparameters minimizing overfitting, not the training loss.

Justification of transforming the task into transfer learning problem that utilizes fine-tuning with different tasks is as follow: Just transfer learning is like sequentially addressing diverse tasks during passing down pre-trained parameters2.3, imposing severely heterogeneous regularization can be regarded as transfer learning because it not only serves the heterogeneous task on each period, which can be supported by the drastic increase of training loss3, but also sustains the identical set of parameters, being trained within each heterogeneous tasks. Also, it can be confirmed that the consumed time needed to reach for certain training loss gradually decrease 2.3, which means that the parameters trained in previous periods can facilitate the parameters to be trained on the heterogeneous task. Therefore, periodically imposing highly heterogeneous regularization can be regarded as transfer learning task.

Thus periodic regularization can leverage learning techniques such as RL or fine-tuning. As examples of task transformation and leveraging the learning techniques, subsequent sections introduce how the original task can be transformed into a different task. Furthermore, how earning methods such as RL or fine-tuning can be implemented into periodic noise injection and periodic dropout is also presented [1.

### 3.4 PERIODIC REGULARIZATION IMPLEMENTING REINFORCEMENT LEARNING

#### 3.4.1 TASK TRANSFORMATION

To implement the RLTuggener et al. (2019), the task that trains the child model Bello et al. (2017)with periodic regularization should be formulated as MDP problem3.3.3. This formulation can be defined as following setting: 1) the state $(S_t)$ should be defined as the parameter or information indirectly related with the parameter to reflect the status of the model. 2) Heterogeneous regularization performed during the transition between adjacent periods can make different loss landscape??, leading to an exclusive trajectory of the parameter $(S_{t+1})$. Therefore, the selecting hyperparameters can be defined as action $(A_t)$ Wu et al. (2020) because the next state $(S_{t+1})$ can be different, depending on the hyperparameter $(A_t)$. 3) Depending on the combination of parameters $(S_t)$ and certain hyperparameters $(A_t)$, the degree of overfitting can vary. Thus, the level of overfitting $(R_t)$ measured by difference between mean of test F1 score and training F1 score can be defined as the reward.

Let $\lambda_{period}$, $HS$, $D$, $Overfit$, $M$ and $S$ denote, respectively, hyperparameter for regularization on each period, hyperparameter space Wu et al. (2020), dataset, level of overfitting, model being trained and state defined by status of model. The goal of this MDP is:

$$\lambda_{period}^* = \underset{\lambda_{period} \in HS}{\arg\min} \, E_{(D_{\text{train}}, D_{\text{valid}}) \sim D}[Overfit(M, S, D_{\text{train}}, D_{\text{valid}}, \lambda_{period})]$$

#### 3.4.2 IMPLEMENTATION

In the MDP formulation, while it's desirable to use the model's parameters as state, this approach can be inefficient due to the vast number of parameters in recent deep learning models. Instead, metrics from training and validation, such as training loss and F1 score are collected during a period and utilized to indirectly represent the state. Moreover, in the periodic noise injection and periodic dropout, selecting the standard deviation of the distribution utilized for sampling noise and drop rate is configured as action the agent can take.

For detail of the process2, during from one period's start($pre\,Period_t$) to its end($post\,Period_t$), random mini-batches from the training dataset are used for training child model. Moreover, metrics such as training loss and F1 score are collected during this period($Period_t$). These information are used for input $(S_{t+1})$ of agent for deciding the action $(A_{t+1})$. Furthermore, at each $post\,Period_t$, the child model is evaluated by some mini-batches from the validation dataset and the test loss and F1 score are collected.

After the validation, using some mini-batch, is over, agent decides hyperparameters $(A_{t+1})$ for next period $(Period_{t+1})$ based on training loss, training F1 score, test loss and test F1 score$(S_{t+1})$. Additionally, the difference between average of training F1 score and test F1 score is computed for the reward $(R_t)$ at $post\,Period_t$

After processing each period and completing one epoch of training dataset, the child model is assessed by entire validation dataset. This evaluation metric is independent of the agent's training process and only utilized to assess the model's performance under chosen hyperparameters $(A_t)$.

### 3.5 PERIODIC REGULARIZATION IMPLEMENTING FINE-TUNING

#### 3.5.1 TASK TRANSFORMATION

Through periodically adjusting hyperparameters making highly heterogeneous regularization3, it is like to train parameters in diverse task3.3.3. Therefore, the original task can be transformed into a transfer learning problem.

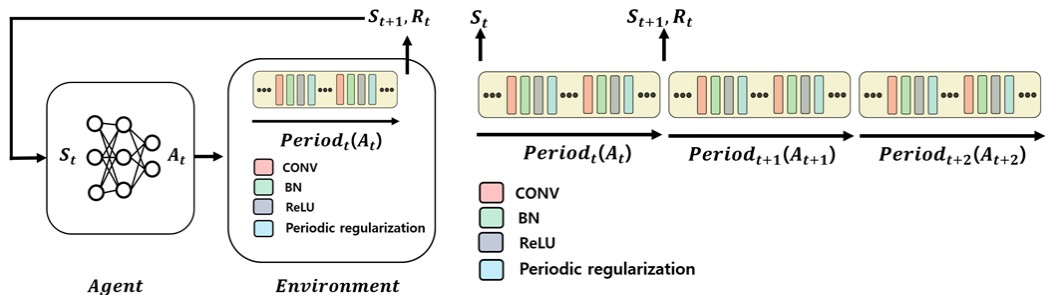

Figure 2: Description of overall process of MDP, the $Period_t(A_t)$ in above figure means that the hyperparameters($A_t$) selected by agent are utilized for regularization during the $Period_t$

### 3.5.2 IMPLEMENTATION

First, a method to change the loss landscape into a very heterogeneous landscape should be devised. Thus, we propose a method called periodic sign changer. This method can be applied to the any neural network having successive modules of batch normalization and the ReLU because the key idea of this approach is the sign change between those layers3. According to the flag periodically toggled with Bernoulli distribution($p = 0.5$), whether the sign change occurs or not is decided and the sign of feature after batch normalization layer is reversed if the flag is true3. This can alter the previous landscape into severely heterogeneous loss landscape 3, which can be confirmed in that there is drastic increases of training loss 3. With sign changer3, repeating the toggling over periods can exploit advantages of transfer learning by periodically fine-tuning the parameters 3 on heterogeneous tasks. This is called Periodically Fine-tuned Regularization(PFR).

### 3.5.3 PFR WITH EXPONENTIALLY WEIGHTED AVERAGE

However, because the parameter is trained with a big length period, there can be a problem of knowledge-vanishing phenomenon where the knowledge the parameters learned from previous periods can perish. Thus Exponentially Weighted Averages (EWA) is mounted on PFR3. This accumulatively add the parameters across periods ($\alpha = 0.9$). This approach is called Periodically Fined-tuned Regularization with Exponentially Wegithed Average (PFREWA)

$$W_{PrePeriod_{t+1}} = \alpha * W_{PostPeriod_t} + (1 - \alpha) * W_{PrePeriod_t} 2$$

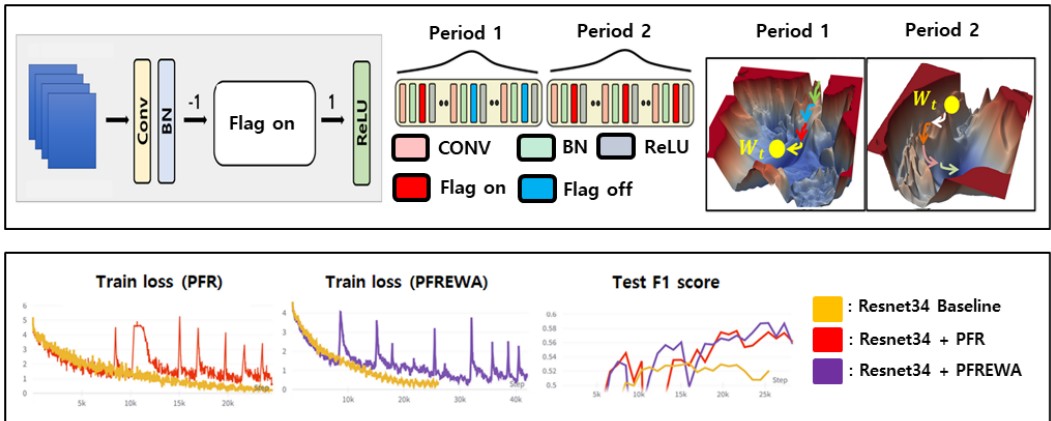

Figure 3: (Above) From left, Reversed sign with Flag on, Dynamically toggled flag along to period and Loss landscape on adjacent periods, (Below) From left, soaring training loss graph with PFR, PFREWA and Test F1 score on the CIFAR100 1, **??**

# 4 EXPERIMENTS AND RESULTS

## 4.1 TRAINING DETAILS

Entire experiments are conducted by utilizing the Pytorch. In the training setting, across the all dataset, the CrossEntrophy is used as a loss function and the Adam optimizer with a weight decay 0.0001. However, the learning rate scheduler is not utilized at all. In the experiments with CIFAR100 and FER 2013 datasets, the batch size and learning rate are, respectively, 64 and 0.001 while 224 and 0.0001 in the Affect Net. In the validation setting, the F1 score is utilized for the metric. In the case of leveraging the RL, the PPO algorithm is utilized for training the agent network. As for hardware setting, All the experiments were conducted using a GPU server equipped with two NVIDIA RTX 3090 GPUs, 128 GB RAM, and an Intel i9-10940X CPU.

| F1 score | BL | GNI | D | PNI | PD | PNIRL | PDRL | PFR | PFREWA |
|----------|------|------|------|------|------|-------|--------|------|--------|
| CIFAR100,Resnet34 | 0.531 | 0.553 | 0.552 | 0.561 | 0.57 | 0.603 | **0.643** | 0.584 | 0.598 |
| FER2013,Resnet18 | 0.615 | 0.602 | 0.601 | 0.612 | 0.605 | 0.616 | **0.626** | 0.620 | 0.624 |
| AffectNet,Resnet34 | 0.482 | 0.480 | 0.486 | 0.475 | 0.487 | 0.492 | **0.534** | 0.482 | 0.494 |

Table 1: Test F1 score according to dataset and model trained aligned regularization, BL=Baseline, GNI=Gaussian Noise Injection, D=Dropout, PNIRL=Periodic Noise Injection with Reinforcement Learning, PDRL=Periodic Dropout with Reinforcement Learning

## 4.2 PERIODIC REGULARIZATION

In the experiments on the CIFAR100 dataset, there is enhanced performance with the application of noise injection and dropout in comparison to the baseline[1]. Furthermore, periodic noise injection and periodic dropout outperform their vanilla counterparts in terms of F1 scores[1], showcasing the potential benefits of extending vanilla regularization with the periodic approach.

Contrastingly, for the FER2013 dataset, all techniques, including vanilla and periodic versions of noise injection and dropout, underperformed relative to the baseline. This is assumed to be because of the nature of the FER domain[1], Li & Deng (2020).

Turning to the Affect Net results, although vanilla dropout and periodic dropout outperform the baseline, the gap isn't significant[1]. Furthermore, both noise injection and periodic noise injection align closely with baseline performance[1]. Much like the FER2013 dataset, these trends may be attributed to the dataset's inherent properties.

## 4.3 PERIODIC REGULARIZATION IMPLEMENTING REINFORCEMENT LEARNING

As the most significant result across entire experiments, if periodic dropout leverages RL, the model performance can be achieved consistently and significantly across all experiments without a single exception[1]. Especially, in experiments on the Affect Net dataset, very monumental results are observed. Although it has been proven that performance improvement due to the nature of the FER domain is really hard, a very remarkable performance improvement can be observed with PDRL[1].

In addition, through result from regularization leveraging the RL, the training loss does not become too low and the test loss does not increase after the minimum point of the test loss graph[4] even though the learning rate scheduler is not utilized at all. This may be attributed to the agent deciding hyperparameters that can minimize overfitting based on the state of the parameters, rather than randomly determining the hyperparameters of each period.

Through the experimental results of the FEF2013 dataset, considering that most regularizations actually worsened the model's performance, it is significant that PDRL succeeds in improving performance[1] even though the performance improvements is slight.

## 4.4 PERIODIC REGULARIZATION IMPLEMENTING FINE-TUNING

As a result of the experiment, PFR (Periodically Fine-tuned Regularization) can improve performance except for the Affect Net dataset. In addition, although the performance improvement was

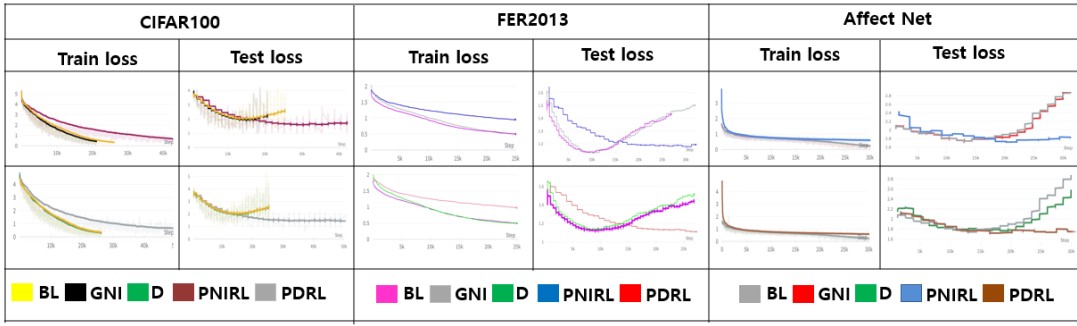

Figure 4: Training loss and test loss according to dataset

slight for the FER2013 dataset, it is a meaningful result considering that most regularizations actually worsened the model's performance1.

In the case of PFREWA, it can be confirmed through experiments that performance can be improved for all datasets. However, the degree of improvement on the Affect Net dataset is trivial compared to periodic dropout implementing RL. Therefore, considering the practicality of the application, it can be said that this method is not a meaningful approach at least in the FER domain.

### 4.5 REPRESENTATION ANALYSIS BY GRAD CAM

To evaluate representation capacity beyond mere numerical metric, we employ grad CAM5, especially spotlighting the impressive improvement of the model trained using periodic dropout with RL1, and contrasting it with other methods.

Starting with the first row of images, the BL4 model exhibits overfitting, which can be confirmed from the result where it focuses on facial outlines, unrelated to emotions. On the other hand, PNIRL concentrate on the forehead, chin, and mouth-adjacent muscles, related to emotions. Furthermore, PDRL also concentrates on the eyes, compared to PNIRL. In the second row, PDRL exclusively and uniquely focuses on mouth muscles, salient indicative of sadness, possibly explaining its superior performance. By the third row, both BL and PNIRL models seem distracted by noise because they also are paying attention on emotion-irrelevant areas(e.g., bubbles around the head). However, PDRL exclusively concentrate on emotion-relevant areas.

Based on the analysis of representation capacity, it can be confirmed that the PDRL can facilitate the model to represent the essential features related to the class even for data with a lot of noise.

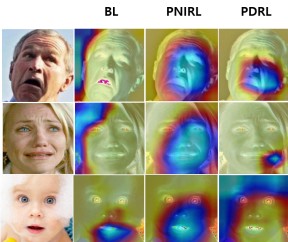

Figure 5: Images applied with grad CAM, degree of attention (Red=high, Yellow=middle, Blue=low)

## 5 CONCLUSION

In conclusion, this paper introduces concepts that can both consistently and remarkably improve the performance of the deep learning model across datasets. In addition, it is meaningful in that traditional regularization can also benefit from integrating concepts such as "periodic regularization", alongside learning methods like "RL" and "fine-tuning" by "task transformation". Especialy, utilizing periodic dropout with RL on challenging datasets such as Affect Net boosts performance, enhancing feature representation, which is significant accomplishment with regard to the characteristic of domain. Therefore, it can be expected that the proposed method will be able to make a significant contribution too in different domains where the problem of overfitting due to noise is severe. While our experiments only focused on regularizations such as noise injection and dropout and learning techniques like RL and fine-tuning, the framework of this research paper can be extensively reproducible on other vanilla regularization. Therefore, Extended research will explore diverse combinations to further improve model performance in challenging domains.

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

# A  APPENDIX

