# OpenReview forum: "Heterogeneity of Regularization between adjacent periods"
_ICLR.cc/2024/Conference — ICLR 2024 Conference Withdrawn Submission_

### Official Review · Reviewer_kYa6 · 2023-10-28

**Soundness:** 1 poor
**Presentation:** 2 fair
**Contribution:** 1 poor
**Rating:** 3
**Confidence:** 5

**Summary:**

In summary, this paper introduces a new regularization method called Periodic Regularization, which involves adjusting hyperparameters periodically during training. When combined with Reinforcement Learning and other techniques, the authors show experimental results in improving performance, particularly in tasks like Facial Expression Recognition. The main goal is to advance research in regularization techniques and deep learning.

**Strengths:**

The idea of periodic regularization is interesting and could potentially benefit deep learning optimization. It can also generalize by adopting existing and future regularization methods into its pipeline, and reducing the manual effort in applying regularization methods in real-world applications.

The paper is well-structured and the interpretation is clear.

**Weaknesses:**

Methodology:
There has been a lot of prior work on meta-learning, especially meta-RL. Given that the regularization parameters (e.g., dropout rate) are usually seen as hyperparameters, the authors should explain the connection between the proposed method and existing meta-learning methods such as MAML, meta-gradients, etc.
In addition, there are also variants of regularization methods that are trained along with the network, e.g., Adaptive Dropout. The authors should mention the literature and explain the differences and novelty.

Experiments:
The main issue with the paper is in experiments. First, the method is only tested on 3 small datasets with 1 ResNet each. It's hard to demonstrate the usability of the proposed method in today's systems. The authors should include more architectures (EfficientNet, ViT, etc.) and experiment on a larger dataset (ImageNet).
Secondly, the reported results are not convincing and actually contradict existing work. For example, for ResNet34 on CIFAR-100, one should easily get >.75 accuracy with open-source implementations (ref: https://huggingface.co/edadaltocg/resnet34_cifar100,  https://github.com/weiaicunzai/pytorch-cifar100). The reported scores are way below the baseline, so it is hard to say if the proposed method actually improves the performance.
Finally, the authors should compare to other methods such as MAML, adaptive dropout, etc.

**Questions:**

See weakness. My main question is why the results are not consistent with existing work.

---

### Official Review · Reviewer_yqeu · 2023-10-30

**Soundness:** 1 poor
**Presentation:** 1 poor
**Contribution:** 1 poor
**Rating:** 1
**Confidence:** 5

**Summary:**

The paper introduces Periodic Regularization, a technique that incorporates periodicity into the dynamic adjustment of regularization hyperparameters during training. By framing the hyperparameter tuning as a reinforcement learning or transfer learning problem, this approach allows for the application of techniques such as RL and fine-tuning to enhance performance. Through experiments, the paper shows that combining Periodic Regularization with RL consistently improves accuracy across a wide range of datasets, sometimes surpassing the performance of conventional regularization methods.

**Strengths:**

The paper considers a crucial problem in deep learning to dynamically adjust hyperparameters over periods during training.

**Weaknesses:**

The paper has many typos, format issues and grammatical mistakes, so the quality is quite poor. For example, 1) there is no space between a word and a reference in many sentences, 2) As *an* representative example..., 3) there is ?? in the caption of Fig. 3.

All figures are low-resolution and hard to read.

**Questions:**

Please improve the quality of the paper.

---

### Official Review · Reviewer_KrfM · 2023-10-31

**Soundness:** 2 fair
**Presentation:** 1 poor
**Contribution:** 1 poor
**Rating:** 1
**Confidence:** 5

**Summary:**

This paper proposes to periodically change the hyperparameters of the deep learning models as a regularization training strategy. Apply this strategy to some regularization techniques, such as dropout and Gaussian noise injection. The author validates the strategy on three datasets: CIFAR100, FER2013 and AFFECTNet.

**Strengths:**

1. The paper's idea is reasonable to some extent. Drawing inspiration from prior research, it suggests that adjusting hyperparameters after a set period, defined by a certain number of forward propagations, is more effective than either consistently maintaining the same hyperparameters or adjusting them during every propagation.
2. The reported results can, to some extent, validate the proposed training technique.

**Weaknesses:**

1. The writing quality of this paper is subpar and hard to follow. Throughout, there are numerous grammatical, typographical, and formatting errors. In the reviewer's opinion, the manuscript does not meet the basic standards for academic writing, let alone the acceptance standards for the ICLR conference.

2. The proposal of the idea lacks theoretical or substantial experimental backing. The introduced concept of periodically updating parameters seems more like a training trick. The explanation and motivation for the idea are weak.

3. The abstract contains numerous formatting errors. There should be a space between the reference and the text, and generally, an abstract should avoid excessive citations.

4. The introduction is ambiguous. Many abbreviations aren't explained in full upon their first appearance but are clarified later. The logic is muddled, and it doesn't offer clear insights or explanations for the proposed strategy.

5. The related work section is too brief and doesn't adequately review research relevant to this paper or highlight distinctions. There's no comprehensive overview; datasets are included under related work, and even GRAD-CAM is treated as related work, despite this paper only using Grad-CAM for feature visualization without modifying it. How is this considered related work?

6. Figure 1 is ambiguous and doesn't positively contribute to clarifying the paper's motivation.

7. In section 2.2, the authors state, "In this paper, R(sk) is denoted as the difference between the mean of test F1 score and training F1 score of the kth sample and b." This suggests that test data has been used during training. The reviewer believes the reported results are not credible, violating the principle that test data should not be used for training.

8. The experimental design, validation, settings, and process descriptions are unclear. Comparisons with state-of-the-art models are missing. The datasets used are mostly toy datasets. There's a lack of validation on large-scale datasets, recent deep models, other machine learning tasks, and ablation studies.

9. The figures presented in the paper are of low resolution, with incorrect font styles. The captions are unclear, and many abbreviations are undefined, as seen in Figures 1, 2, 3, 4, and 5.

10. To the reviewer, this paper neither resembles a rigorous academic manuscript nor a neatly formatted technical report. It appears more like a hastily prepared undergraduate course report. Therefore, the reviewer believes such papers should be desk rejected to avoid burdening the review process further.

**Questions:**

Please refer to the Weaknesses section.

---

### Official Review · Reviewer_QKyW · 2023-11-06

**Soundness:** 3 good
**Presentation:** 2 fair
**Contribution:** 3 good
**Rating:** 5
**Confidence:** 3

**Summary:**

The paper introduces Periodic Regularization, a novel regularization technique that incorporates periodicity into the dynamic hyperparameter tuning. The paper suggests combining Periodic Regularization with other learning techniques such as Reinforcement Learning (RL) and Transfer Learning to further improve model performance. The authors demonstrate the effectiveness of Periodic Regularization in addressing the overfitting problem in Facial Expression Recognition (FER) tasks, where traditional regularization techniques have proven ineffective. The authors argue that Periodic Regularization can enhance model performance across different datasets and automatically adjust hyperparameters to prevent overfitting.

**Strengths:**

+ This paper shows that periodic regularization can improve models using different learning frameworks such as reinforcement learning and transfer learning.
+ The paper experiments show that Periodic Regularization can effectively solve the over-fitting problem in the FER task.
+ Periodic regularization can automatically adjust hyperparameters.
+ The idea is great and works well in practice.

**Weaknesses:**

+ The motivation is unclear. Heterogeneity is not well defined. It is confusing that we should "introduce heterogeneity in the hidden layer to enhance the adaptability and generalization of the model". Then, the author only explains how PR is designed, but ignores why it is designed this way.
+ Writing needs improvement. Please do not abuse table/picture hyperlinks in your paper.

**Questions:**

1. Could you further explain what the heterogeneity is (by some examples)? To better understand the motivation.

2. One of the advantages of PR is that it can automatically adjust hyperparameters. However, PFR also incorporates another hyperparameter $\alpha$ in the equation of Section 3.5.3. And the sensitivity of this hyperparameter is not given. How is the empirical value $\alpha=0.9$ chosen?

---

### Meta-Review · Area_Chair_u4vW · 2023-11-28

**Metareview:**

This paper was poorly written and its clarity is indeed a major issue. Our reviewers consistently agreed to reject it.

**Justification For Why Not Higher Score:**

This paper was poorly written and its clarity is indeed a major issue. Our reviewers consistently agreed to reject it.

**Justification For Why Not Lower Score:**

N/A

---

### Decision · Program_Chairs · 2024-01-16

Reject